# Serological Prevalence of and Risk Factors for *Coxiella burnetti* Infection in Women of Punjab Province, Pakistan

**DOI:** 10.3390/ijerph19084576

**Published:** 2022-04-11

**Authors:** Shahzad Ali, Usama Saeed, Muhammad Rizwan, Hosny El-Adawy, Katja Mertens-Scholz, Heinrich Neubauer

**Affiliations:** 1Wildlife Epidemiology and Molecular Microbiology Laboratory (One Health Research Group), Discipline of Zoology, Department of Wildlife & Ecology, University of Veterinary and Animal Sciences, Lahore, Ravi Campus, Pattoki 55300, Pakistan; usama.saeed@uvas.edu.pk (U.S.); rizwanasif400@gmail.com (M.R.); 2Friedrich-Loeffler-Institute, Institute of Bacterial Infections and Zoonoses, Naumburger Str. 96a, 07743 Jena, Germany; hosny.eladawy@fli.de (H.E.-A.); katja.mertens-scholz@fli.de (K.M.-S.); heinrich.neubauer@fli.de (H.N.); 3Faculty Medicine of Veterinary, Kafrelsheikh University, Kafr El-Sheikh 33516, Egypt

**Keywords:** Q-fever, risk assessment, women, pregnancy, Pakistan

## Abstract

Background: *Coxiella burnetii*, the etiological agent of Q (query) fever, provokes abortions in ruminants and is suspected to cause adverse pregnancy outcomes in women. Infection of pregnant women is linked with high mortality and morbidity of the fetus and the mother is at high risk to acquire chronic Q fever. This research was conducted to evaluate the prevalence of Q fever in women and to detect associated risk factors in four districts of Punjab Province, Pakistan. Methods: A total of 297 blood samples were obtained from 147 pregnant and 150 non-pregnant women of the districts Okara, Jhang, Chiniot and Faisalabad of Punjab, Pakistan. Data related to risk factors and demographic parameters were collected using a questionnaire. Serum samples were screened for phase I and phase II specific IgG antibodies for antigens of phase I and phase II using ELISA tests. Univariate and binary regression were used to analyze important risk factors of Q fever. Results: Twenty-five serum samples (8.4%) were found seropositive for Q fever. Seventeen women were positive for Phase-I and twenty-one were positive for phase-II antibodies. Highest and statistically significant (*p* < 0.05) seroprevalence of 17.1% was observed in Faisalabad. Age, urbanicity, living status, pregnancy status, abortion history, occupation, and consumption of tap water were positively correlated (*p* < 0.05) with Q fever, while being aged, urbanity, low income, contact with animals and consumption of tap water was identified as potential risk factors. Conclusions: Q fever is prevalent in women of Pakistan. There is a need for an awareness program about the importance of *C. burnetii* infections and prevention strategies in women during pregnancy to minimize adverse pregnancy outcomes.

## 1. Introduction

Human population is at high risk of acquiring emerging infectious diseases particularly those of zoonotic nature [1]. Among zoonotic diseases, Q fever is of great significance with special reference to human public health. The causative agent of Q fever is *Coxiella* (*C*.) *burnetii* which is a Gram-negative bacterium [2]. The transmission of infection to human beings occurs through direct and indirect routes. Ruminants are considered as the main reservoir for human infections [3,4]. In these animals an infection is often asymptomatic but can lead to abortions and weak offspring. The bacteria are shed in urine, feces, milk and in tremendously amounts within birth products [5,6]. The direct routes of transmission of infection from infected animals to humans are contact with unattended birth products and body fluids [7]. 

Consumption of unpasteurized milk and its products is at least associated with a seroconversion [3,8,9]. The most common indirect source of infection are aerosols from infected farm animals because *C. burnetii* can remain in the environment over long periods of time and is transported by winds over long distances [10,11,12]. Other animals such as cats, dogs, rabbits, wild animals and birds have also been described as hosts [13,14,15,16,17,18]. In urban settings, outbreaks of Q fever in humans were linked with serological-positive pet cats [19].

Acute Q fever can be symptomatic or asymptomatic in human beings. The clinical conditions of an acute infection with *C. burnetii* are diverse: hepatitis, atypical pneumonia or flu-like illness [15,16,17,18]. Moreover, clinical outcomes of patients with acute infection vary from country to country (i.e., The Netherlands, Spain, France, Kenya) which includes abdominal pain, cough, chest pain, diarrhoea, dyspnoea, fatigue, fever, headache, joint pain, muscle pain, night sweating, nausea, and vomiting [3,20,21,22,23,24,25,26,27]. Chronic Q fever may develop from an acute infection. Possible predisposing factors are preexisting vascular grafts, cardiac valvulopathy, immunosuppression, and aneurysms. However, a combination of serological testing and clinical presentation helps in accurate identification of chronic human Q fever cases [28,29].

The incidence of adverse pregnancy outcomes (APOs) (both acute and persistent infection) has been investigated using seropositivity as marker and was inconsistently associated with low birth weight, preterm birth, congenital malformations, abortions or fetal death [30,31,32,33]. Most people of Pakistan live in rural areas. These under-privileged persons mainly depend on livestock for livelihood and are often involved in food production (i.e., milk, butter, meat, etc.). Women play an important role in the management of livestock at the small household level. 

Only few studies related to Q fever seroprevalence in the human population have been conducted previously in Pakistan. A seroprevalence of 10.19% phase-I and 11.8% phase-II was reported using microtiter complement fixation in humans of Northern Pakistan [34]. However, 26.8% individuals were found positive for Q fever by complement fixation test in Karachi, Sindh, Pakistan [35]. The present study was conducted for serological detection of a *C. burnetii* infection and risk factors for women of four districts of Punjab Province, Pakistan. 

## 2. Materials and Methods

### 2.1. Sampling Area

The four districts Chiniot, Okara, Faisalabad and Jhang of Punjab were included in the study. District Chiniot is famous for its agricultural production, and it is located on the left side of the Chenab River. Jhang is located at the junction of the two rivers Chenab and Jhelum. Okara city is well known for its military dairy farms and cotton mills, and it is situated adjacent the river Ravi. District Faisalabad is the third largest city of Pakistan, and it is famous all around the world for its textile industry. The people of these districts live in urban and rural areas. Livestock keeping is an important source of food and income of the rural communities. Management of livestock is done by women at household level especially in rural areas. The important animal species of these districts are cattle, buffaloes, sheep and goats [36]. According to the 2017 census, the total number of the human population of all four districts was 15,026,205 [37]. 

### 2.2. Data Collection 

This cross-sectional study was designed with no follow-up investigation. Data related to demography and epidemiology were collected on blood sampling day using a questionnaire covering location/district, urbanicity, living status, pregnancy status, abortion history, occupation, age, contact with animals (livestock), consumption of raw milk, and consumption of tap water.

#### 2.2.1. Collection of Blood Sample and Processing

A total of 297 blood samples were obtained from women (147 pregnant and 150 non-pregnant). In total, 4 mL of blood was collected from each woman using sterile syringes and transferred to blood collection tubes (non-EDTA). These samples were centrifuged at 3000 rpm for 5 min for separation. The sera were collected in 1.5 mL reaction tubes and kept at −40 °C for further analysis. 

#### 2.2.2. Serological and Interpretation

Serum samples were examined for IgG antibodies using phase I and phase II serion ELISA kits (Virion\Serion, Würzburg, Germany) and assessed according to the manufacturer’s instructions. Briefly, for phase-I, sera with an optical density (OD) > 10% above the OD from the cut off samples were recorded as positive, those with ODs < 10% below the OD from the cut off sera were recorded as negative. Samples between these values were considered as the borderline. For phase II, the cut off range was determined on the base of the standard curve fixed by the mean of the extinction of the standard serum as per the instructions of manufacturers. An outcome of <20 U/mL was considered as negative, 20–30 U/mL borderline, whereas >30 U/mL was considered positive [3].

### 2.3. Statistical Analysis 

For the statistical analysis, individuals were considered seropositive based on parallel interpretation of phase-I and phase-II ELISA results. Chi-square test was performed to detect the significance of association between seropositive samples. The significance level (*p*-value) was ≤0.05%. Multivariable logistic regression was used to determine, corresponding 95% CIs, odds ratios (ORs) and *p*-values to analyze the relationship between the demographic data, risk factors and seropositivity by using statistical software (SPSS Version 21.0) [38].

## 3. Results

Twenty-five serum samples (8.4%) were found seropositive for Q fever based on the parallel interpretation of the two ELISA tests (Table 1). Among these, 17 (5.7%) individuals were seropositive for Q fever in phase-I, while 21 (7.07%) were seropositive for Q fever in phase-II. The highest seroprevalence (17.1%) was documented in Faisalabad district as compared to Chiniot (7.7%), Okara (5.6%), and Jhang (2.8%). This finding was significant (*p* = 0.01) (Table 1).

The seroprevalence of Q fever was higher in women from urban areas (12.4%) when compared to women from rural areas (5%). The seroprevalence of Q fever was much higher in women with lower living status (13.7%). This variation was statistically (*p* = 0.026) significant. The prevalence of *C. burnetii* was 11.6% (*n* = 17) in pregnant women and 5.3% (*n* = 8) in non-pregnant. Seroprevalence of Q fever appeared to be higher in those females that had a history of abortion (12.2%) while lower in females without a history of abortion (5.1%).

The prevalence of Q fever could be correlated with women’s professions. Those women who were livestock farmers and had direct contact with animals or animal samples showed a higher (18.5%; *n* = 12) seroprevalence for Q fever when compared to housewives (7.6%; *n* = 5) and teachers (5.3%; *n* = 3) (Table 1). The prevalence of *C. burnetii* was 8.9% in those who had close contact with animals and 7.9% in those who had no direct contact with animals. A higher seroprevalence was shown for women who reported to not consume raw milk (10.2%) as compared to those who consume raw milk (5.4%). Seroprevalence of *C. burnetii* was greater (11.9%) in those women with access to tap water if compared to those who consumed drinking water from sewerage channels and/or stagnant water (4.3%). In terms of age groups, 19.5% positive samples were found in the age group ≥ 41 years. Seroprevalence was 6.2%, 5.4% and 2.5% for women of age groups ≤ 20, 31–40 and 21–30 years, respectively. These data show that age group was statistically significant for prevalence’s of Q fever (*p* = 0.001) (Table 2).

Based on binary logistic regression, being aged, urbanity, low-income status, contact with animals and consumption of tap water are potential risk factors (*p* < 0.05) for the seropositivity while other risk factors were found non-significant for this setting (*p* > 0.05) (Table 3).

## 4. Discussion

Q fever caused by *Coxiella burnetii* is a notorious zoonosis with its main reservoirs in sheep, goats and cattle. Humans are infected via inhalation contaminated aerosols, dust particles or close contact with placentas and birth products of often symptomless animals. These materials can carry bacteria in high numbers and one particle may already cause disease. In humans main risk factors for developing Q fever are cardiac diseases, pregnancy and aneurysms [28]. Vaccines are available for animals. Q fever vaccination is recommended for people in high risk professions, but can cause severe side effects [39,40,41].

From Pakistan, two studies older than 20 years on prevalence for Q fever in men are available from Karachi city of Sindh province and from the Southern and in Northern areas with prevalences of 26.8% and 11.8%, respectively [34,35]. In the present study, an 8.4% seroprevalence was found in women living in four districts of the Punjab. It has to be stressed that the studies should be compared with care. Besides the different diagnostic methods used, the different study groups and the changes in animal production have to be taken in account. In the Northern area high risk professionals (i.e., livestock farmers, veterinarians, laboratory animal handers) and patients with clinical signs of pyrexia of unknown origin and cardiac diseases were investigated. In Karachi city, blood samples of patients with typhoid fever, enteric fever and pneumonia were screened for Q fever antibodies. In this study, pregnant and non-pregnant women from with and without contact with animals were investigated. It can be assumed that in this group the seroprevalence should be lower than in women from rural areas who are exposed to household dairy animals on a daily basis. This assumption is corroborated by findings from rural Iranian areas where a prevalence of 29.3% was reported from pregnant women in steady contact to farm animals [42]. In areas with no animal contact such as in London, or with indirect contact only, such as in The Netherlands, seroprevalences of 4.6% and 9% were recorded [32,43].

The prevalence of Q fever in women ranged from 17.1% in district Faisalabad to 7.7%, 5.6% and 2.8% in districts Chiniot, Okara and Jhang, respectively. These findings are in line with the results of previous study on seroprevalence of zoonotic disease such as *Brucella abortus* in pregnant women from Rawalpindi [44]. The presence of Q fever in small ruminants in the Punjab Pakistan including Okara is well documented [45]. Thus, this existing animal reservoir is expected source of the transmission to humans. The prevalence of *C. burnetti* DNA in soil samples was successfully linked to seroconversion of small ruminants in a previous study conducted in different districts of the Punjab, including Faisalabad [46]. The presence of *Coxiella* bacteria in the soil is considered a threat for the human population of that area [47].

In the selected study area, women from low-income families were more often seropositive (13.7%) than those from families with medium and high living standards (3.6–6.2%). Quite an opposite situation was observed in a previous study in which women from middle standard families were significantly more often seropositive (54.0%) when compared to women from low- and high-income families in Turkey [48]. The result of this study is in agreement with that of a study on brucellosis in which women from families with lower income had also a high risk of disease in Iran [49]. This might be due to the fact that both diseases (Q fever, brucellosis) are of zoonotic nature and poor women handling animals usually have no adequate access to personal protection and medical care due to poverty. Hence, it is a well-known fact that zoonotic diseases are more prevalent in low-income families in general. Hence, the results from different countries should always be compared with care. 

*Coxiella burnetii* can cause complications (i.e., premature birth, abortions, intrauterine growth retardation, etc.) in pregnant women and their unborn children. Thus, well-timed diagnosis of disease can be important to avoid pregnancy associated complications [45]. Our study revealed that prevalence of Q fever was higher (12.2%) in women who had a history of abortion as compared to those who did not (5.1%). These findings are comparable to the results of Abushahba et al., [50] who also described a higher seroprevalence in women who had an abortion history (32.2%). A much higher prevalence was also reported in women with abnormal pregnancy history (39.8%) and a lower one in women with normal pregnancy (23.8%) in Parsabad and Ahvaz regions of Iran [42]. Pregnant women may also be more prone to infection as we found a higher prevalence of Q fever in pregnant woman (11.6%) when compared with a non-pregnant woman (5.3%). Hence, this assumption needs further research to be proven. However, antibodies against *C. burnetii* during pregnancy do not necessarily mean that *C. burnetii* is sole cause of reproductive abnormalities. Infection with other pathogens associated with animal contacts such as brucellae and *Toxoplasma gondii* can be possible confounders [49]. 

In previous studies conducted in Ilam province of Iran (37.93%) and in Egypt (46.15%) the seroprevalences of Q fever were found to be much higher in women with animal contact than in this study [50,51]. Hence, the variable “contact with animals” was proved by binary logistic regression as a potential risk factor for Q fever (95% CI = 0.052–0.762, OR = 0.199, *p* = 0.02) in this study. The possible reason is that most women shared their living environments with animals. Moreover, poor farmers cannot afford to cull or dispose diseased animals which are often then the source of the continuing spread of diseases. It is well known from many studies that the most frequent vehicle for transmission are aerosols caused by infected farm animals. *C. burnetii* can survive in the environment over long periods and is transmitted by winds over long distances [10,11,12]. Thus, it poses a permanent threat for the human population of Punjab, Pakistan [44].

In this study not all seropositive women were rearing animals or had contact with animals. However, occupational risk of infection had a great impact on the seroprevalence of Q fever. Women working as livestock farmers had a higher seroprevalence (18.5%) than others, i.e., housewives (7.6%), teachers (5.3%), businesswomen (4.8%) or students (4.5%). Even in highly industrialized countries such as The Netherlands, 3.2% seroprevalence of Q fever was observed in women of the agriculture sector [26] but only 0.5% women of the meat processing industry including abattoirs and 0.7% of the general population were positive. The observed elevated seroprevalence in our study can possibly be linked to the time of exposure, poor handling techniques and missing protective means while dealing with infected animals. 

The prevalence of Q fever in this study was higher in women who did not consume raw milk (9.3%) than in those who consumed raw milk (6.2%). In contrast, a significant relationship was detected in consumption of raw milk (*p* < 0.05) and seropositivity for Q fever in Jordan [52]. Consumption of unpasteurized cheese and raw milk was related to Q fever with human disease in United Kingdom also. The local geographical situation may have contributed to our findings, e.g., that herds were grazed hillside and main wind direction was directed towards near cities as transmission via consumption of contaminated products is less likely than transmission via contaminated aerosols from infected livestock or their parturient products [9]. 

*C. burnetti* DNA was detected from sewage water and wastewater of Q-fever-positive goat farms which may had been the source of environmental contamination in The Netherlands [53]. Moreover, detection of *C. burnetii* in river water from Rome suggested that contaminated water can be a contributing factor to Q fever seropositivity in humans [54]. A low level of risk was determined for transmission of *C. burnetti* via inhalation of drinking water aerosols during showering in a previous study [55]. Consequently, the role of some additional factors like contaminated tap water, raw vegetables and fruits as possible sources for *C. burnetti* were investigated in this study. Indeed, a higher prevalence was observed in those women who consumed tap water (11.9%) when compared with those who took water from other sources (4.3%). Hence, these findings are not conclusive and need further studies as tap water use simply may reflect poor living conditions of women of the agriculture sector of Punjab.

Among different age groups, women > 40 years were more at risk (19.5%) to bear anti-*Coxiella* antibodies. These results are comparable to that of previous study conducted in Kurdistan, Iran [49]. Therefore, the chances of infection with pathogens or contact with their antigens may increase with age because of the longer exposure times due to traditionally sharing of the living environments with animals. However, the present study results are not comparable with a study conducted in Switzerland in which the age group < 15 years were at high risk of infection with Q fever [56].

Finally, phase-specific serology allows differentiation of an acute or recent Q fever infection from chronic disease. During an acute infection, antibodies against phase II antigens are predominant with IgG levels higher than IgM. Chronic Q fever is indicated by a high IgG titer to phase I antigen [57]. There were used ELISA tests allowed only qualitative interpretation of the results for IgG against phase II and quantitative results for IgG against phase I. Therefore, positive serological results indicate exposure to *C. burnetii*, but does not allow differentiation of an acute, past or chronic infection. Additional testing for IgM would have been beneficial, since IgM antibodies against phase II are detectable during the acute phase of Q fever.

## 5. Conclusions

These data indicate that a zoonotic disease such as Q fever may become a serious threat to public health and especially to pregnant women, when there is a lack of control strategies. An educational program related to pregnancy and the risk of zoonotic diseases, and their prevention is needed. For pregnant women with close contact to animals and signs of an infectious disease, a routine serological examination covering zoonotic diseases including Q fever should be made available in Pakistan. Moreover, there is a need for a more general control strategy for Q fever in animals involving the use of a vaccine to prevent spread of Q fever from animals to humans. Foodborne contamination of *C. burnetii* can be minimized by the proper boiling of water and milk before human use in remote settings.

## Figures and Tables

**Table 1 ijerph-19-04576-t001:** Seroprevalence of *C. burnetii* antibodies based on epidemiological and demographic factors in women of Punjab, Pakistan, by using Chi-square analysis.

Category	Variable	Examined	Positive	Prevalence (%)	X^2^	*p*-Value
District	Okara	71	4	5.6	11.179	0.01
	Jhang	72	2	2.8		
	Chiniot	78	6	7.7		
	Faisalabad	76	13	17.1		
Urbanicity	Urban	137	17	12.4	5.255	0.022
	Rural	160	8	5		
Living status	High	97	6	6.2	7.3	0.026
	Medium	83	3	3.6		
	Low	117	16	13.7		
Pregnancy status	Pregnant	147	17	11.6	3.739	0.05
	Non-pregnant	150	8	5.3		
Abortion history	Yes	139	17	12.2	4.92	0.03
	No	158	8	5.1		
Occupation	Housewife	66	5	7.6	11.380	0.023
	Teacher	57	3	5.3		
	Students	67	3	4.5		
	Livestock Farmer	65	12	18.5		
	Business Women	42	2	4.8		
Contact with animal (livestock *)	Yes	157	14	8.9	0.108	0.7
	No	140	11	7.9		
Consumption of raw milk	Yes	111	6	5.4	2.086	0.149
	No	186	19	10.2		
Consumption of tap water	Yes	159	19	11.9	5.538	0.02
	No	138	6	4.3		

Livestock * = sheep, goat, cattle, buffalo.

**Table 2 ijerph-19-04576-t002:** Seroprevalence of *C. burnetii* antibodies in different age group of women of Punjab, Pakistan by using Chi-square analysis.

Category	Variables	Examined	Seropositive	Prevalence (%)	X^2^	*p*-Value
Age Group	≤20	65	4	6.2	17.246	0.001
	21–30	81	2	2.5		
	31–40	74	4	5.4		
	≥41	77	15	19.5		

**Table 3 ijerph-19-04576-t003:** Identification of Risk factors associated with Q fever based on seroprevalence in women from four districts of Punjab, Pakistan, by using binary logistic regression.

Risk Factors	Variable	OR	SE	95% CI	*p*-Value
Age	≤20	1.908	0.829	0.376–9.697	0.436
	21–30	14.375	0.953	2.220–93.071	0.005
	31–40	12.255	0.912	2.051–73.221	0.006
	≥41	R			
Location	Okara	4.9	0.890	0.857–28.024	0.07
	Jhang	17.94	1.027	2.396–134.33	0.005
	Chiniot	4.9	0.762	1.102–21.84	0.037
	Faisalabad	R			
Urbanicity	Urban	7.002	0.696	1.788–27.415	0.005
	Rural	R			
Living status	High	1.332	0.845	0.254–6.975	0.7
	Medium	8.920	0.825	1.769–44.979	0.008
	Low	R			
Pregnancy status	Pregnant	0.393	0.774	0.086–1.792	0.2
	Non-Pregnant	R			
Previous abortion	Yes	0.514	0.688	0.133–1.980	0.3
	No	R			
Occupation	Housewife	0.874	1.158	0.090–8.465	0.908
	Teacher	0.607	1.254	0.052–7.095	0.691
	Students	3.235	1.464	0.184–57.017	0.423
	Livestock Farmer	0.501	1.132	0.054–4.611	0.542
	Business women	R			
Contact with animal (livestock)	Yes	0.199	0.686	0.052–0.762	0.02
	No	R			
Consumption of raw milk	Yes	2.442	0.692	0.629–9.480	0.2
	No	R			
Consumption of tap water	Yes	0.220	0.683	0.058–0.840	0.02
	No	R			

R—Reference Value; SE—Standard Error; CI—Confidence Interval; OR—Odd Ratio.

## Data Availability

The data presented in this study are available within the article.

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
