# Peer review of "Serological Prevalence of and Risk Factors for Coxiella burnetti Infection in Women of Punjab Province, Pakistan"

_ijerph, 2022, doi:10.3390/ijerph19084576_

Round 1
Reviewer 1 Report
It is very nice manuscript, I found only one mistake. Table 1 consumption of tap water includes double line with No, in my oppinion is mistake because total number is 297, so line yes is 159 and above line NO 138 is 297.Author Response
Please see the attachment

Reviewer 2 Report
This research "Serological prevalence of and risk factors for Coxiella burnetti infection in women of Punjab Province, Pakistan" was conducted to evaluate the prevalence of Q fever in women and to detect associated risk factors in four districts of Punjab Province, Pakistan", for it, a total of 297 blood samples were obtained from 147 pregnant and 150 non-pregnant women of the districts Okara, Jhang, Chiniot and Faisalabad of Punjab, Pakistan. Data related to risk factors and demographic parameters were collected a using questionnaire, also, serum samples were screened for specific IgG antibodies of phase I and phase II using ELISA tests.
This is an interesting article, however, it is necessary to consider some aspects
Edit and italicize “Coxiella burnetii” throughout the manuscript.
Introduction
Lines 48-49: Reference 13 is very limited in relation to what is being referred to, please include more citations, especially those related to wildlife and that are more recent, such as “González-Barrio D, Ruiz-Fons F. Coxiella burnetii in wild mammals: A systematic review. Transbound Emerg Dis. 2019;66(2):662-671. doi: 10.1111/tbed.13085”.
Line 57: order these references from lowest to highest
Materials and methods
Data collection
Lines 92-95: here are some risk factors that do not appear in Table 1 of results.
On the other hand, there are some closely related risk factors, such as occupation (e.g. being a farmer) and contact with animals. Normally, if you are a farmer you will be in contact with production animals, whether they are cows, sheep, goats, among others.
Regarding the variable "contact with animals", the type of animals should be specified, whether they are production animals (related to the occupation), pets or wildlife.
Results
Table 1 and 2: Please change the title of the table and make it more coherent with what appears in the table.
Table 1: In “consumption of tap water”, modify the column of bussinees women since there are two "no".
Lines 150-152: Please keep the formatting of the text, these lines appear in a smaller size.
Discusion
The discussion should be re-written and restructured in order to better understand the paper and the results. In my opinion, it is very extensive and sometimes several aspects are confused during its reading.
Line 174: change “of Iran by [38]” to “of Iran by Ullah et al., [38].
Lines 174-176: Clarify this affirmation, it is not clear
Line 214: change “results of [45]” to “results of Abushahba et al., [38]
Lines 219-242: this section should be clarified, since the risk of occupation (livestock farmer) is related to exposure to production animals, and the discussion contradicts this. In my opinion, it would be necessary to differentiate the type of animal contacted, for example, it could be that pets (dogs and cats) could be a risk factor, since urban women have a higher seroprevalence.
Rewrite this part of the discussion and make changes in the description of the risk factors related to contact with animals.
Reviewer 3 Report
The manuscript investigated the prevalence of and risk factors for Coxiella burnetti infection in women in Punjab province. The study is well described and well written and it worths to be published.
Few main concerns:
In my opinion, the IgG antibodies test is good as screening test but I suggest authors to add in the discussion that It would be good to consider even IgM towards phase I and II to confirm positive results.
Round 2
Reviewer 2 Report
This revised version of the article offers a more readable version.
I have no additional comments.